# Impact of perioperative organ injury on morbidity and mortality in 28 million surgical patients

Felix Kork [1,7] ✉, Yafen Liang [2,3,7] ✉, Adit A. Ginde [4], Xiaoyi Yuan [2], Rolf Rossaint[1], Hongfang Liu [5], Alex S. Evers[6] & Holger K. Eltzschig [2,3] ✉

Perioperative organ injury contributes to morbidity and mortality of surgical patients. This cohort study included all elective and emergent surgeries in Germany over 4 years to address the impact of perioperative organ injuries on outcomes. We analyzed 28,350,953 cases. In-hospital mortality was 1.4% (*n* = 393,157), and 4.4% of cases (*n* = 1,245,898) experienced perioperative organ injury. Perioperative organ injury was associated with 9-fold higher odds of death and prolonged hospital stay by 11.2 days. Acute kidney injury had the highest incidence (2.0%) and was associated with 25.0% mortality. While delirium had the second highest incidence (1.5%), it was associated with the lowest mortality (10.8%). This was followed by acute myocardial infarction (incidence 0.6%, mortality 15.6%), stroke (incidence 0.6%, mortality 13.1%), pulmonary embolism (incidence 0.3%, mortality 20.0%), liver injury (incidence 0.1%, mortality 68.7%), and acute respiratory distress syndrome (incidence 0.1%, mortality 44.7%). These findings help prioritize interventions for preventing or treating individual types of perioperative organ injury.

Despite advances in perioperative care, significant morbidity and mortality still occur following surgery and remain a major public health burden[1–3]. A recent modeling study estimated that about 234 million major surgeries are performed globally every year[4], and depending on the country and institution, up to 4% of patients are predicted to die during that hospitalization. In addition, up to 15% will experience severe postoperative morbidity[5–8], thereby amounting to an estimated 1000 deaths and 4000 major complications every hour[5]. The perioperative period has been referred to as "the neglected stepchild of global health"[9], and therapeutic strategies to improve perioperative outcomes are areas of intensive investigation[2,9,10].

Previous studies indicate that acute perioperative organ injury causing single- or multiple-organ failure remains a leading cause of morbidity and mortality in surgical patients[3,11,12]. Perioperative organ injury encompasses different types of acute dysfunction of individual organs, which can be triggered by an inflammatory response to a "perioperative stressor"[11]. These stressors include direct surgical stimulation during the operation or indirect stimuli such as blood transfusions or cardio-pulmonary bypass, or an underlying disease that has triggered the surgical intervention (such as an infection)[13–15]. Perioperative organ injury can be exacerbated by comorbidities, concurrent hypoxia, hypoperfusion, or ischemia-reperfusion injury and modulated by genotypic susceptibility[9,16]. It can affect essentially every organ system, with delirium, stroke, acute myocardial infarction (AMI), acute respiratory distress syndrome (ARDS), pulmonary embolism (PE), liver injury (LI), and acute kidney injury (AKI) being

[1]Department of Anesthesiology, Medical Faculty, RWTH Aachen University, Aachen, Germany. [2]Department of Anesthesiology, Critical Care and Pain Medicine, McGovern Medical School, the University of Texas Health Science Center at Houston, Houston, TX, USA. [3]Center for OUTCOMES RESEARCH and Department of Anesthesiology, UTHealth, Houston, TX, USA. [4]Department of Emergency Medicine, University of Colorado School of Medicine, Aurora, CO, USA. [5]Department of Health Data Science and Artificial Intelligence, McWilliams School of Biomedical Informatics, the University of Texas Health Science Center at Houston, Houston, TX, USA. [6]Department of Anesthesiology, Washington University, School of Medicine in St. Louis, St. Louis, MO, USA. [7]These authors contributed equally: Felix Kork, Yafen Liang. ✉e-mail: fkork@ukaachen.de; yafen.liang@uth.tmc.edu; holger.eltzschig@uth.tmc.edu

the most commonly identified types of perioperative organ injury[17–23].

Prior studies have looked at the impact of individual organ injury on surgical mortality and morbidity in small groups of specific patient populations[17–23]. However, the global implications of perioperative organ injury and the potentiating effect of multiple perioperative organ injuries have not been comprehensively investigated in a large cohort of surgical patients. Moreover, there is a lack of studies addressing the relative contributions of individual organ injuries to perioperative outcomes. Therefore, it has been challenging to prioritize targeted interventions to improve this pressing global health problem. To address this knowledge gap, we performed a population-based, retrospective cohort study to determine the relative incidence and impact of perioperative organ injuries in all hospitalized surgical patients between 2014 and 2017 in Germany.

## Results

### Study population

28,350,953 surgeries conducted in Germany between 2014 and 2017 were included in the analysis. In Germany, documentation of every inpatient episode is legally required. Therefore, missing data are very rare. We identified 693 cases with missing information on sex or age that we excluded from the analysis. In-hospital mortality was 1.4% ($n = 393,157$). The cohort's median age was 59 years; 54.3% of them were of female gender, and 29.2% were admitted as emergencies. Most patients were healthy with no comorbidities. 33.7% of patients underwent high-risk surgery (Tables 1 and S1). Patients who experienced perioperative organ injury were older, more often of male gender, had more comorbidity burden, and more frequently underwent emergent or high-risk surgery. For perioperative organ injury and surgical outcomes categorized by type of surgery, please refer to Table S2.

### Perioperative organ injury

4.4% ($n = 1,245,898$) of cases experienced perioperative organ injury during hospitalization. This included neurologic injury (delirium: 1.5% and stroke: 0.6%), AMI (0.6%), ARDS (0.1%), PE (0.3%), LI (0.1%), and AKI (2.0%; Fig. 1, Tables 2 and S3). Patients who were diagnosed with perioperative organ injury had higher mortality rates (17.0% vs. 0.7%, $P < 0.001$) and longer hospital length of stay (HLOS) (17 vs. 4 days, $P < 0.001$) compared to patients without such a diagnosis (Table 2 and Fig. S1). In an adjusted analysis, any perioperative organ injury was associated with a 9-fold higher odds of death (OR = 9.3, 95% CI = [9.2, 9.4]) and a prolonged HLOS (beta = 11.2 days, 95% CI = [11.1, 11.2]; Tables 3 and S4). Additional adjusted analysis for time to in-hospital death and time to hospital discharge confirmed these results. Patients who experienced perioperative organ injury had a higher risk of death (HR = 3.51, 95% CI= [3.49, 3.54], $P < 0.001$) and lower chance of hospital discharge (HR = 0.45, 95% CI= [0.45, 0.45], $P < 0.001$; Table S4). Among surgical patients who died during hospitalization, 54% ($n = 211,584$) experienced perioperative organ injuries.

### Multiple perioperative organ injury

Among all patients, 3.7% experienced one organ injury, 0.6% experienced two organ injuries, and 0.1% experienced three or more organ injuries perioperatively. Increased numbers of organ injuries were associated with significantly increased mortality (mortality rate of 13.9%, 31.0%, and 43.7%, respectively, for 1, 2, or 3 or more organ injuries compared with a mortality rate of 0.7% for no organ injury, $P < 0.001$) and HLOS (HLOS of 16, 24, 29 days respectively for 1, 2, or 3 or more organ injuries compared with HLOS of 4 days for no organ injury, $P < 0.001$; Fig. 2 and Table S5). In a risk-adjusted analysis, increased numbers of perioperative organ injuries were associated with exponentially increasing odds of death (Table 3). One organ injury was associated with a 7-fold increased odds of death and a longer

HLOS of 10 days compared to patients without organ injury. Two concomitant organ injuries were associated with a 20-fold increased odds of death and a longer HLOS of 18 days. Three or more concomitant organ injuries were associated with a 40-fold increased odds of death and a longer HLOS of 24 days (Tables 3, S6). Additional adjusted analysis for time to in-hospital death showed similarly increased risks of death and decreased chances of hospital discharge for multiple organ injuries (Tables 3, S6).

### Specific organ injuries

Among all organ injuries included in this study, AKI represented the most frequent type of perioperative organ injury (2.0%), followed by delirium (1.5%), AMI (0.6%), stroke (0.6%), PE (0.3%), LI (0.1%) and ARDS (0.1%) (Table 2, Fig. 1). The mortality associated with different types of perioperative organ injury is as follows: LI (68.7%), ARDS (44.7%), AKI (25.0%), PE (20.0%), AMI (15.6%), stroke (13.1%), delirium (10.8%) (Table 2, Fig. 1, Figs. S2–S8). Delirium is the second most frequent type of organ injury but is associated with the lowest mortality. ARDS and LI are the least common organ injuries but are associated with the highest mortality. The mortality rate of specific organ injuries is also different when they present themselves as a single organ injury vs. one of the multiple organ injuries. For example, delirium was associated with the lowest mortality rate as a single organ injury (6.6%), but the mortality rate increased 3 times when it occurred with other organ injuries (21.6%) (Fig. 1). Adjusted analysis showed LI had the highest increased odds of death, followed by ARDS, AKI, PE, AMI, stroke, and delirium (Table 3).

In addition to mortality, it is important to identify each organ injury's morbidity burden. We pursued this by analyzing HLOS data. Among all perioperative organ injuries, ARDS was associated with the longest HLOS, followed by LI, AKI, delirium, AMI, PE, and stroke (Table 2, Figs. S2–S8). Adjusted analysis showed ARDS prolonged HLOS by the most, followed by delirium, AKI, PE, stroke, AMI, and LI (Tables 3 and S7, for sensitivity analyses, see Tables S8–S10). Patients with ARDS frequently require mechanical ventilation and intensive care unit admission, which could significantly increase healthcare costs compared with other organ injuries that may not require mechanical ventilation or intensive care unit stay (such as AKI).

Our results demonstrated that AKI is the largest contributor to perioperative death, followed by delirium, AMI, and liver injury (Fig. S9). Risk factors associated with different types of organ injury are presented in Table S11.

## Discussion

Perioperative organ injury contributes substantially to surgical mortality and morbidity[1,24–26]. In this population-based, retrospective cohort study, we determined in-hospital mortality of 1.4% and incidence of perioperative organ injury of 4.4% in over 28 million surgeries performed in Germany. Additionally, any perioperative organ injury increased the odds of death by 9-fold and prolonged HLOS by 11 days. Increased numbers of perioperative organ injuries were associated with an exponentially increased risk of death. A similar impact was observed for HLOS as a morbidity burden. Among different organ injuries, AKI occurred most frequently and had the largest impact on perioperative death, while liver injury was associated with the highest mortality rate.

Perioperative death represents a significant burden to global healthcare. A recent analysis using England's combined Hospital Episode Statistics and Office of National Statistics (HES-ONS) dataset suggested that 4.2 million people worldwide die within 30 days of surgery each year[27]. This accounts for 7.7% of all deaths globally, making it the third largest contributor to deaths after ischemic heart disease and stroke. The current study showed that 393,157 surgical patients died perioperatively in Germany between 2014 and 2017. If perioperative death could be considered a single separate disease

**Table 1 | Characteristics of the study cohort categorized by perioperative organ injury**

| Characteristic | All patients (N = 28,350,953) | Patients with POI‡ (N = 1,245,898) | Patients without POI (N = 27,105,055) |
|---|---|---|---|
| Median age (IQR)—yrs | 59 [39–74] | 76 [66–82] | 58 [38–73] |
| Female sex—no. (%) | 15,390,796 (54.3) | 516,057 (41.4) | 14,874,739 (54.9) |
| Hospital admission type—no. (%) | | | |
| Referral from physician | 19,309,039 (68.1) | 384,096 (30.8) | 18,924,943 (69.8) |
| Emergency | 8,287,300 (29.2) | 681,935 (54.7) | 7,605,365 (28.1) |
| Transfer from another hospital | 733,057 (2.6) | 179,105 (14.4) | 553,952 (2.0) |
| Transfer from rehabilitation | 2188 (<0.1) | 383 (<0.1) | 1805 (<0.1) |
| Birth | 19,369 (<0.1) | 379 (<0.1) | 18,990 (<0.1) |
| Median Charlson comorbidity index (IQR)—pts | 0 [0–2] | 3 [1–5] | 0 [0–1] |
| *Charlson Comorbidity Index Items—no. (%)* | | | |
| Diabetes mellitus | 3,843,335 (13.6) | 392,551 (31.5) | 3,450,784 (12.7) |
| Uncomplicated | 2,952,801 (10.4) | 258,435 (20.7) | 2,694,366 (9.9) |
| With end-organ damage | 890,534 (3.1) | 134,116 (10.8) | 756,418 (2.8) |
| Cancer | 3,156,980 (11.1) | 221,567 (17.8) | 2,935,413 (10.8) |
| Non-metastatic | 2,246,180 (7.9) | 133,818 (10.7) | 2,112,362 (7.8) |
| Metastatic | 910,800 (3.2) | 87,749 (7.0) | 823,051 (3.0) |
| Renal disease | 2,336,360 (8.2) | 406,537 (32.6) | 1,929,823 (7.1) |
| Congestive heart failure | 1,910,719 (6.7) | 427,143 (34.3) | 1,483,576 (5.5) |
| Chronic obstructive pulmonary disease | 1,750,439 (6.2) | 169,875 (13.6) | 1,580,564 (5.8) |
| Peripheral vascular disease | 1,670,506 (5.9) | 227,379 (18.3) | 1,443,127 (5.3) |
| Cerebrovascular disease | 812,635 (2.9) | 177,533 (14.2) | 635,102 (2.3) |
| History of myocardial infarction | 599,614 (2.1) | 67,993 (5.5) | 527,621 (1.9) |
| Dementia | 634,339 (2.2) | 126,623 (10.2) | 507,716 (1.9) |
| Hemiplegia or paraplegia | 574,671 (2.0) | 160,401 (12.9) | 414,270 (1.5) |
| Liver disease | 546,908 (1.9) | 79,929 (6.4) | 466,979 (1.7) |
| Mild | 418,008 (1.5) | 52,203 (4.2) | 365,805 (1.3) |
| Moderate to severe | 128,900 (0.5) | 27,726 (2.2) | 101,174 (0.4) |
| Rheumatoid disease | 292,112 (1.0) | 21,018 (1.7) | 271,094 (1.0) |
| Peptic ulcer disease | 230,826 (0.8) | 53,472 (4.3) | 177,354 (0.7) |
| AIDS | 9179 (<0.1) | 934 (<0.1) | 8236 (<0.1) |
| High-risk surgeries—no. (%)* | 9,546,235 (33.7) | 848,557 (68.1) | 8,697,678 (32.1) |
| Abdominal surgery | 6,596,343 (23.3) | 407,507 (33.7) | 6,188,836 (22.8) |
| Thoracic surgery† | 1,530,479 (5.4) | 184,536 (14.8) | 1,345,943 (5.0) |
| Cardiac surgery | 1,101,545 (3.9) | 207,308 (16.6) | 894,237 (3.3) |
| Intracranial surgery | 304,022 (1.1) | 46,128 (3.7) | 257,894 (1.0) |
| Transplantation surgery | 13,846 (<0.1) | 3078 (0.2) | 10,768 (<0.1) |

*High-risk surgeries included abdominal surgery, thoracic surgery, cardiac surgery, intracranial surgery, and transplantation surgery.
†Thoracic surgery does not include cardiac surgery.
‡Any kind of perioperative organ injury.
*POI* perioperative organ injury, *AIDS* autoimmune deficiency syndrome.

entity regardless of the causes for surgery (such as cancer and acute myocardial infarction that could have led to surgery and the patient could die perioperatively from cancer or acute myocardial infarction), this would account for the third leading cause of death in Germany (Fig. S10, Panel A), following circulatory system diseases and neoplasms. The number of perioperative deaths in Germany would extrapolate to an annual death of approximately 383,000 in the USA, making perioperative death the third leading cause of death in the USA as well (Fig. S10, Panel B). To put the magnitude of this health burden into the context of the COVID-19 pandemic, the estimated burden of perioperative deaths would approximate 68% of the total deaths from COVID-19 during the first year of the pandemic (560,115 deaths from COVID-19 by 9 April 2021, in the USA)[28]. Therefore, perioperative death is a pressing global healthcare burden of a very large magnitude.

A critical step to improve perioperative outcomes is identifying key contributors to perioperative death and prioritizing prevention or treatment efforts. Our study showed about 46% of patients who died perioperatively did not experience any organ injury. The reasons could be two-fold: first, the nature of administrative data might have allowed the detection and recording of more severe forms of organ injury, and some of the milder forms of organ injury may not be recorded. Secondarily, other etiologies of perioperative death, e.g., surgical complications such as acute massive hemorrhage or infection, are not captured in our study since our focus was perioperative organ injury.

The present study found that AKI was the most common type of organ injury and was associated with the highest number of perioperative deaths. The incidence and impact of AKI on the surgical outcome are mostly consistent with previous study findings[29,30]. However, compared to some previous studies, the incidence of mild AKI might be disproportionally low compared to the more severe form of AKI. The overall impact of AKI on mortality might be overestimated because more severe forms of AKI are more likely captured[31].

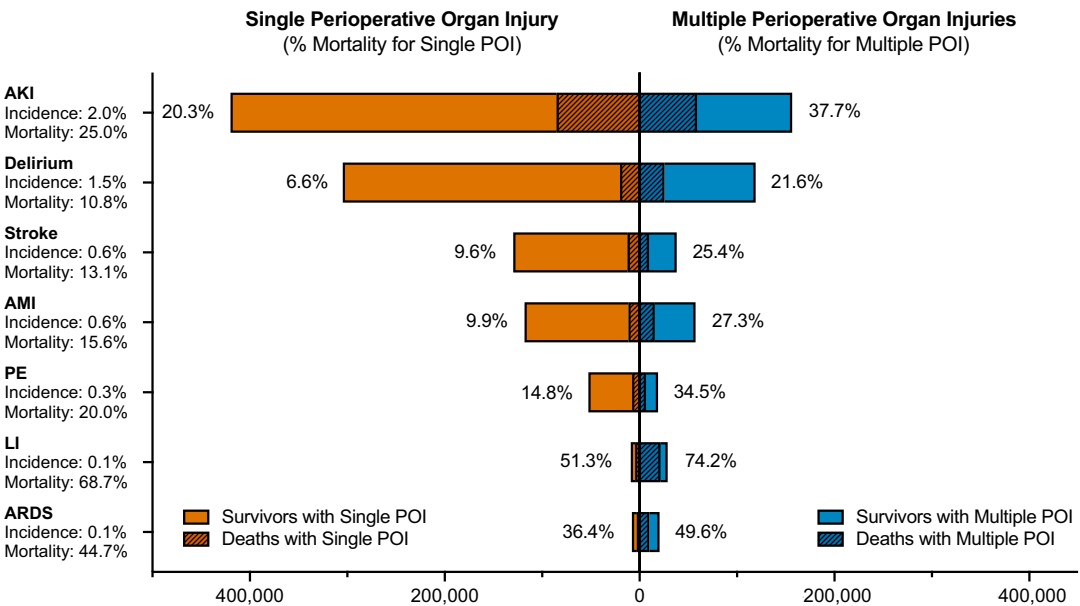

**Fig. 1 | Incidence and mortality of perioperative organ injury.** Perioperative organ injuries included delirium, stroke, acute myocardial infarction (AMI), acute respiratory distress syndrome (ARDS), pulmonary embolism (PE), liver injury (LI), and acute kidney injury (AKI). Each organ injury is separated into its impact as an individual organ (left panel side) or as concomitant with additional other organ injuries (right panel side). For example, AKI has a mortality rate of 20.3% when it occurs as an isolated organ injury (left panel side). However, if it happens in conjunction with another organ injury, AKI-associated mortality increases to 37.7% (right panel side). Note: among all the organ injuries investigated, AKI was the most common, while Acute Liver Injury was associated with the highest mortality.

**Table 2 | Outcome of patients with and without perioperative organ injury**

| Any perioperative organ injury | Any POI (N = 1,245,898) | No POI (N = 27,105,055) |
|---|---|---|
| Median HLOS (IQR)—d | 17 [10–29] | 4 [2–8] |
| In-hospital mortality—no. (%) | 211,584 (17.0) | 181,573 (0.7) |
| **Perioperative delirium** | **Delirium (N = 423,649)** | **No Delirium (N = 27,927,304)** |
| Median HLOS (IQR)—d | 19 [11–31] | 4 [2–8] |
| In-hospital mortality—no. (%) | 45,876 (10.8) | 347,281 (1.2) |
| **Perioperative stroke** | **Stroke (N = 167,695)** | **No Stroke (N = 28,183,258)** |
| Median HLOS (IQR)—d | 17 [10–29] | 4 [2–9] |
| In-hospital mortality—no. (%) | 22,041 (13.1) | 371,116 (1.3) |
| **Perioperative AMI** | **AMI (N = 175,556)** | **No AMI (N = 28,175,397)** |
| Median HLOS (IQR)—d | 15 [9–25] | 4 [2–9] |
| In-hospital mortality—no. (%) | 27,359 (15.6) | 365,798 (1.3) |
| **Perioperative ARDS** | **ARDS (N = 29,312)** | **No ARDS (N = 28,321,641)** |
| Median HLOS (IQR)—d | 29 [16–48] | 4 [2–9] |
| In-hospital mortality—no. (%) | 13,092 (44.7) | 380,065 (1.3) |
| **Perioperative PE** | **PE (N = 71,633)** | **No PE (N = 28,279,320)** |
| Median HLOS (IQR)—d | 17 [10–30] | 4 [2–9] |
| In-hospital mortality—no. (%) | 14,359 (20.0) | 378,798 (1.3) |
| **Perioperative LI** | **LI (N = 37,961)** | **No LI (N = 28,312,992)** |
| Median HLOS (IQR)—d | 19 [9–35] | 4 [2–9] |
| In-hospital mortality—no. (%) | 26,089 (68.7) | 367,068 (1.4) |
| **Perioperative AKI** | **AKI (N = 576,438)** | **No AKI (N = 27,774,515)** |
| Median HLOS (IQR)—d | 19 [11–32] | 4 [2–8] |
| In-hospital mortality—no. (%) | 144,306 (25.0) | 248,851 (0.9) |

*AMI* acute myocardial infarction, *ARDS* adult respiratory distress syndrome, *PE* pulmonary embolism, *LI* acute liver injury, *AKI* acute kidney injury, *SIRS* systemic inflammatory response syndrome, *POI* perioperative organ injury, *HLOS* hospital length of stay, *IQR* interquartile range.

Additionally, it is important to note that attributable fractions are also based on the assumption of a causal relationship between risk factors and outcomes. The interaction and overlapping between different types of perioperative organ injury could not be further delineated using this method. Nonetheless, some studies suggest that even minor subclinical postoperative kidney dysfunction, which does not qualify for AKI can impact surgical outcomes. For example, a recent study in 39,000 surgical patients demonstrated that minor postoperative

**Table 3 | Risk-adjusted associations of in-hospital mortality and morbidity with any perioperative organ injury, type of perioperative organ injury, and number of perioperative organ injuries from multivariable regression analysis models**

| | Mortality* | | Morbidity[†] | |
|---|---|---|---|---|
| | **In-hospital death Odds ratio (95% CI)** | **Survival Hazard ratio (95% CI)** | **Hospital length of stay Coefficient (95% CI)** | **Discharge Hazard ratio (95%CI)** |
| Perioperative organ injury[‡] | | | | |
| Any organ injury | 9.28 (9.22–9.35) | 3.51 (3.49–3.54) | 11.15 (11.11–11.19) | 0.45 (0.45–0.45) |
| Type of perioperative organ injury[‡] | | | | |
| Delirium | 1.40 (1.38–1.42) | 0.81 (0.80–0.82) | 10.59 (10.53–10.66) | 0.54 (0.54–0.54) |
| Stroke | 2.99 (2.93–3.05) | 1.56 (1.54–1.58) | 6.47 (6.35–6.59) | 0.70 (0.70–0.71) |
| AMI | 3.28 (3.23–3.34) | 1.82 (1.80–1.85) | 4.28 (4.20–4.37) | 0.65 (0.65–0.65) |
| ARDS | 10.60 (10.28–10.94) | 1.75 (1.80–1.85) | 16.04 (15.67–16.41) | 0.32 (0.31–0.32) |
| PE | 5.27 (5.15–5.39) | 1.88 (1.85–1.92) | 9.01 (8.85–9.17) | 0.48 (0.47–0.48) |
| LI | 24.96 (24.28–25.65) | 3.48 (3.43–3.53) | 3.01 (2.71–3.30) | 0.31 (0.30–0.31) |
| AKI | 7.91 (7.84–7.98) | 3.15 (3.12–3.17) | 10.24 (10.18–10.30) | 0.46 (0.46–0.46) |
| Number of perioperative organ injury[‡] | | | | |
| 1 POIs | 7.36 (7.31–7.42) | 3.23 (3.20–3.25) | 9.81 (9.77–9.85) | 0.48 (0.48–0.48) |
| 2 POIs | 19.73 (19.49–19.98) | 4.55 (4.50–4.60) | 17.45 (17.32–17.57) | 0.32 (0.31–0.32) |
| ≥3 POIs | 41.67 (40.58–42.79) | 5.30 (5.20–5.40) | 23.69 (23.32–24.06) | 0.23 (0.23–0.23) |

*Mortality was assessed using three binary logistic regression models estimating the odds ratio and three corresponding Cox proportional hazard regression models estimating the hazard ratio for all-cause in-hospital death, introducing any perioperative organ injury, type of perioperative organ injury or number of perioperative organ injuries, respectively; for the complete models and details see supplementary appendix.

[†]Morbidity was assessed using three robust regression models estimating the coefficients for hospital length of stay and three corresponding Cox proportional hazard models calculating the hazard ratio for hospital discharge, introducing any perioperative organ injury, type of perioperative organ injury or number of perioperative organ injuries, respectively; for the complete models see supplementary appendix.

[‡]odds ratios, hazard ratios and coefficients are adjusted for age, sex, emergency hospital admission, all items of the Charlson comorbidity index, and high-risk surgeries; for the complete models see supplementary appendix.

*AMI* acute myocardial infarction, *ARDS* adult respiratory distress syndrome, *PE* pulmonary embolism, *ALI* acute liver injury, *AKI* acute kidney injury, *SIRS* systemic inflammatory response syndrome, *POI* perioperative organ injury.

creatinine increases were associated with a 2-fold increased risk of death and two days longer HLOS[32]. Therefore, the impact of renal injury may be underestimated in the current study. To conclude, this analysis revealed AKI as the leading contributor to perioperative death, therefore making interventions to prevent or treat perioperative AKI a high priority.

Our study showed perioperative LI is rare (0.1%) but associated with the highest mortality (mortality: 68.7%). Compared to AKI, Perioperative ALI is a much less investigated perioperative organ injury. Mild liver dysfunction in patients without liver disease is common following major surgery[33,34]. However, acute "ischemic hepatitis," a diffuse injury that arises secondary to hypoperfusion (hemodynamic instability) and hypoxia, and is exacerbated by systemic inflammation, could be associated with significant perioperative mortality and morbidity, especially[22] in patients with pre-existing liver disease and cardiac dysfunction. Preventive and therapeutic measures for perioperative ALI are urgently needed.

ARDS was the least frequent (incidence: 0.1%) but the second most deadly type of perioperative organ injury in the current patient cohort (mortality: 44.7%). Moreover, it was associated with the longest HLOS. These findings are consistent with prior study findings[35,36]. Besides supportive care, there is a lack of targeted treatment options[30,37]. Thus, novel therapeutic approaches targeting, for example, hypoxia-inducible transcription factors, adenosine receptors[38], or microRNAs[39] might represent emerging pharmacologic ARDS treatment options to improve patient outcomes[40]. Finally, advancements in our understanding of pathologic and clinical subtypes of ARDS will likely also play a critical role in designing clinical trials to identify the efficacy of individual treatments in specific cohorts of ARDS patients.

The present study found that delirium was the second most frequent (1.5%) perioperative organ injury after AKI. However, it had the lowest mortality rate (10.8%). The mortality rate significantly increased when it occurred together with other organ injuries. These findings

suggest that isolated delirium may not be as harmful as other types of organ injuries, but could be viewed more as a surrogate marker for patient's overall illness[41]. Despite its relatively low mortality, delirium prolonged HLOS by 10.6 days, confirming it is a significant morbidity burden.

Our study results may differ from previous studies regarding the incidence of perioperative organ injury and perioperative mortality rate[42–45]. Additionally, the impact of AMI is less prominent in our study than in previous studies[46,47]. For example, the VISION study by Devereaux et al. investigated the incidence of myocardial injury after noncardiac surgery (MINS) in 21,842 patients undergoing inpatient noncardiac surgery[43]. They found the incidence of MINS to be 17.9% and 30-day mortality to be 1.2% in this cohort. However, it is important to notice that our study represents the complete census of all surgical patients in Germany. In contrast, previous studies, such as the VISION study, focused on older patients or high-risk surgeries[43,44]. The VISION study used troponin biomarker for the diagnosis of myocardial injury, while our study used administrative clinical data to diagnose acute myocardial infarction. Therefore, the endpoint of the study is also different. Additionally, none of the studies compared the relative disease burden of individual organ injury in a single population of surgical patients. As such, they do not provide a basis to prioritize the disease burdens of different organ injuries. They are not designed to guide efforts to prioritize strategies tackling this pressing global health issue. Furthermore, the current study is among the first to demonstrate the interaction between multiple concomitant organ injuries, representing a common problem in surgical patients.

Some important limitations of the current study need to be pointed out. First, the data presented are derived from a single European country, which may not represent other countries or global healthcare systems. However, the current data set is a complete census of all German healthcare data and is highly comprehensive and representative[48,49]. Secondly, organ injury coding may contain

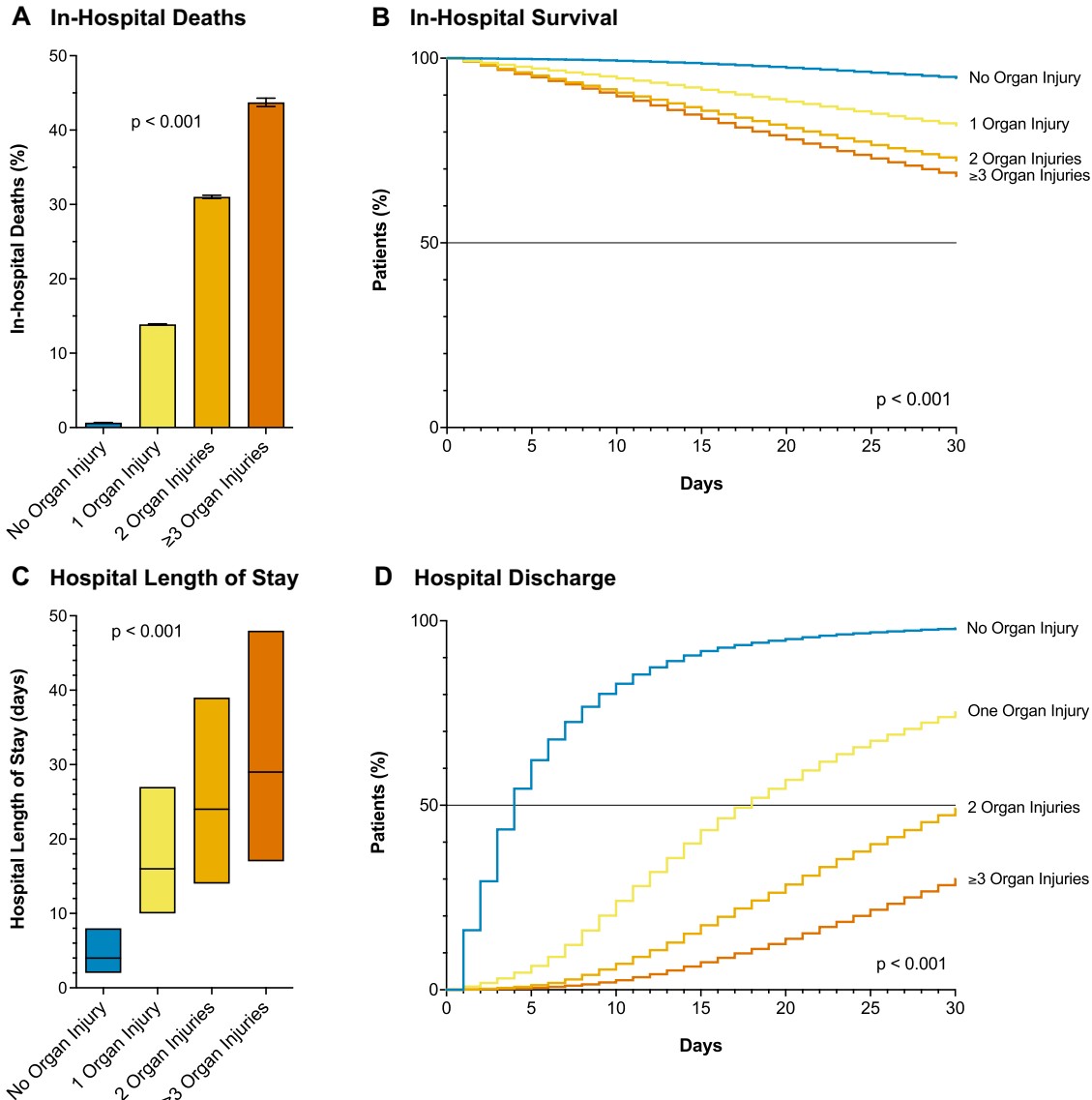

**Fig. 2 | Impact of multiple perioperative organ injuries on surgical mortality and morbidity. A** The mortality rate for patients with no organ injury, 1 or 2, or 3 or more were 0.7%, 13.9%, 31.0%, and 43.7%, respectively (chi-squared test, $p < 0.001$; error bars: 95%CI). **B** Kaplan–Meier survival curves show the probability of 30-day in-hospital survival for patients with different numbers of organ injuries (log-rank test, $P < 0.001$). **C** The hospital length of stay (HLOS) for patients with no organ injury, 1, 2, or 3 or more organ injuries were 4, 16, 24, and 29 days, respectively (Median test, $P < 0.001$, median and quartiles). **D** Kaplan–Meier hazard curves show the probability of 30-day hospital discharge for patients with different numbers of organ injuries (log-rank test, $P < 0.001$). Among all patients, 3.7% ($n = 1,042,713$) experienced one organ injury, 0.6% ($n = 173,959$) experienced two organ injuries, and 0.1% ($n = 29,491$) experienced three or more organ injuries perioperatively.

inaccuracies and be influenced by external coding incentives and other biases. However, in Germany, physicians are legally required to use established guidelines for diagnosing specific types of organ injuries. The Medical Service of Health Funds in Germany regularly audits the coding of hospital discharge diagnoses. Thirdly, it is important to recognize that due to the lack of granularity of the data, we cannot identify subclinical organ injury and its impact, even though other studies have shown that even subclinical organ injuries are closely associated with poor perioperative outcomes[32,43]. Therefore, the impact of perioperative organ injury is likely underestimated in our study even though the impact is already immense. This lack of granularity may also affect our ability to determine the timeline of organ injury in contrast to the index surgery and differentiate pre-existing comorbidities from organ injury after surgery. However, our analysis was performed diligently to ensure that all ICD-codes, including sub-codes, are mutually exclusive in either a perioperative organ injury or

comorbidity category. Fourthly, the current study only included hospitalized surgical patients, and the impact of perioperative organ injury on the out-patient surgical population is not addressed. Similarly, the impact of emergency surgery vs. elective surgery could not be determined in our study due to lack of such data, but investigating the role of emergency surgery on POI and surgical outcomes in this large database would be very insightful as some literature shows that emergent surgery is a significant risk factor for poorer patient outcomes[50,51]. Lastly, multiple studies have shown that in-hospital mortality is an incomplete measure of surgical mortality[52,53], since a significant percentage of procedural deaths occur after transfer or discharge from the index hospital. Since all the patient data are de-identified before being sent to the federal statistics office, we cannot link these data/patients with a death record. Therefore, we cannot provide one-year mortality data. Examing the one-year mortality would be a more comprehensive assessment of the impact of

perioperative organ injury, as has been used in other studies[54]. Similarly, selection bias, such as the impact of HLOS on the capture of perioperative organ injury, lacking of readmission data therefore to link the index surgery to perioperative organ injury that developed after hospital discharge, can not be overlooked. However, our study showed only 0.01% of the cases were re-admissions from post-hospital rehabilitation facilities. Additionally, due to the differences in the healthcare system, HLOS in Germany is much longer than in the U.S., which allows for a more comprehensive capture of POI data.

In summary, the current study demonstrated that perioperative death is a pressing global health burden. Perioperative organ injury was strongly associated with surgical morbidity and mortality. It is important to note that the current study only represents association but does not infer causation. However, the exponentially increased morbidity and mortality with multiple organ injuries is striking and makes measures targeting common pathophysiological pathways very attractive. As most surgeries (more than two-thirds in our cohort) are elective, preventative interventions to reduce perioperative organ injury are vital to improving the surgical outcome.

## Methods
### Data source
In this retrospective cohort study, we accessed and analyzed data from the German Diagnoses Related Groups Statistik (access granted for project # 3333-2017). All hospitalized patients in Germany who underwent surgery and were subsequently discharged or died during their hospital stay between January 1, 2014, and December 31, 2017, were included in the analysis. Each hospitalization is considered as one case for data analysis. Since we used de-identified data and accessed via controlled remote data processing, no Institutional or other Review Board approval was required. Similar studies used the same data access approach without Review Board approval[55]. For details regarding the data source, access, and analysis modalities of the patient sample and variable transcoding, refer to the supplementary appendix (Tables S12 and S13).

### Patient population
The Federal Statistical Office predefined surgical cases as cases with at least one procedure code from Chapter 5 of the OPS (Table S12). These were inpatient surgery cases when patients stayed at least overnight. Therefore, no day surgery, intervention, or diagnostic procedures are included. Anonymous data of all patients treated in German hospitals must be reported to the Institute for the Hospital Reimbursement System by law. These data are then transmitted in accordance with §21 Hospital Reimbursement Act to the German Federal Statistical Office. Since documenting every inpatient episode is mandatory, this study can be considered complete nationwide (excluding military and psychiatric services)[56]. No sample size or power considerations were made. FK coded the analysis protocol, and the Federal Statistical Office performed the analysis.

### Outcomes and exposures
The study's primary endpoint was in-hospital death; the secondary endpoint was hospital length of stay (HLOS). Exposures were perioperative organ injuries, defined as delirium, stroke, AMI, ARDS, PE, LI, or AKI.

### Statistical analysis
We compared categorical variables using the chi-squared test and continuous variables using the Mann–Whitney U test. We fit different regression models to estimate the association of organ injury with in-hospital death and HLOS. For all regression models, we selected dependent variables based on clinical relevance and availability from the database. This included age, sex, emergency admission, comorbidities, and undergoing high-risk surgery. High-risk surgeries are

defined as thoracic surgery (excluding cardiac surgery), cardiac surgery, abdominal surgery, and transplantation surgery. Binary logistic regression models were fit and cross-validated to assess the association of any perioperative organ injury, individual types of organ injuries, and multiple organ injuries with death, and robust regression models for the association with HLOS. When analyzing Kaplan-Meier curves and corresponding Cox proportional hazard models, we considered HLOS as the time to death if we censored patients at in-hospital death or HLOS as the time to discharge if we censored patients at hospital discharge. Additional sensitivity analyses were performed to confirm the associations of organ injuries with HLOS. This included a log-transformed linear model, a competing risk model, and a proportional odds model (Tables S8–S10).

To prioritize intervention efforts to improve surgical outcomes, it is important to identify the relative contribution of individual organ injuries to perioperative mortality. This contribution is determined by calculating the attributable fraction using the following formula: $AFp = (Ip - Iu)/Ip$, while Ip is the incidence in the population, Iu is the incidence in the unexposed group.

We conducted all statistical analyses using Stata Server (Stata 15 for Windows, StataCorp, College Station, TX, USA). We created figures using Prism 9.0.1 for MacOS (GraphPad, San Diego, CA, USA). Given the large sample size, $p < 0.01$ was considered statistically significant. Please refer to the supplementary appendix for a complete description of the methods.

### Reporting summary
Further information on research design is available in the Nature Portfolio Reporting Summary linked to this article.

## Data availability
Due to German privacy laws, individual de-identified participant data cannot be shared directly. Access should be requested from the German Federal Statistical Office: www.destatis.de and through the application portal: www.forschungsdatenzentrum.de/de/antrag. Scientists are granted three years' access to the requested data upon request to answer a specific research question, with the option of a three-year extension. The process of controlled remote analyses is described in detail in the supplementary methods.

## Code availability
The Stata codes used to conduct all analyses for this article are deposited at https://doi.org/10.5281/zenodo.14933270. All analyses can be reproduced with access to the raw files at the German Federal Statistics Office.

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

## Acknowledgements

We would like to acknowledge Melanie Heiliger from the German Federal Statistics Office for her strong support of our project. F.K. is supported by institutional and departmental sources. Y.L. is supported by the Society of Cardiovascular Anesthesia Mid-Career Award, National Institutes of Health (NIH)—Artificial Intelligence/Machine Learning Consortium to Advance Health Equity and Researcher Diversity (AIM-AHEAD) Program and The University of Texas System Trauma Research and Combat Casualty Care Collaborative (TRC4) grant. H.K.E. is supported by National Institute of Health Grants: R01HL154720, R01HL154720-03S1, R01HL165748, R01HL169519, R01DK122796, R35HL177402, T32GM135118 and Department of Defense Grant: W81XWH2110032. X.Y. is supported by the Parker B. Francis Fellowship; American Lung Association Catalyst Award CA-622265, R01HL155950, and R01HL169519. All other authors declare no conflict of interest.

## Author contributions

F.K. performed and analyzed the results for the study. Y.L. interpreted the data and drafted the manuscript. A.A.G. and A.S.E. provided critical advice for the project direction and edited the manuscript. X.Y., H.L. and R.R. edited the manuscript. H.K.E. conceptualized the project, provided critical guidance, and finalized the manuscript.

## Competing interests

Drs. Holger Eltzschig, Xiaoyi Yuan, and Yafen Liang received research funding through a contract between Akebia Therapeutics and UTHealth to support a clinical trial on the effect of vadadustat in hospitalized COVID-19 patients (NCT04478071). Akebia Therapeutics is not involved in conceptualization, design, data collection, analysis, the decision to publish, or the preparation of the manuscript. The remaining authors declare no competing interests.
