## [Transparent Peer Review file · Nature Communications]

Impact of Perioperative Organ Injury on Morbidity and Mortality in 28 Million Surgical Patients

Corresponding Author: Dr Holger Eltzhig

Version 0:

Reviewer comments:

Reviewer #1

(Remarks to the Author)

What are the noteworthy results?

As stated in my previous review [reviewer 1], these are certainly noteworthy given extent and size of database explored.

Will the work be of significance to the field and related fields? How does it compare to the established literature? If the work is not original, please provide relevant references.

The authors have satisfactorily addressed my previous concerns- their response and discussion is measured and an exemplar of the dialogue that should happen in the review process! The issue of "under-diagnosing" is addressed, which remains a significant flaw but is now well explained to non experts and experts alike.

Does the work support the conclusions and claims, or is additional evidence needed?
Conclusions stack up on basis of data presented.

Are there any flaws in the data analysis, interpretation and conclusions? Do these prohibit publication or require revision?
All addressed from Nat Med R1.

Is the methodology sound? Does the work meet the expected standards in your field?
Yes

Is there enough detail provided in the methods for the work to be reproduced?
Code for database should be provided for other researchers to examine/explore.

Reviewer #3

(Remarks to the Author)

Many thanks for the opportunity to look at this updated manuscript. The authors have addressed my concerns and performed significant additional work.

Reviewer #4

(Remarks to the Author)

Thank you for asking me to review this revised manuscript which describes an analysis of administrative care records of patients undergoing surgery in Germany between 2014-2017. The authors report the association between perioperative organ injury and in-hospital outcomes among 28M patients.

There were many excellent points raised by the first round of peer review that

have been generally well addressed. I have a handful of comments based on my own experience of doing large administrative data research, the two most important are:

1. The possibility of significant information bias is not presently addressed.

for example, in Table S3; 27% had Stage 1 AKI and 27% had stage 3 - this is out of keeping with existing descriptions of the distribution of AKI prevalence (e.g. OAKS study: PMID:35029656)

This suggests that patients with less severe organ injury are less likely to be captured (e.g. a KDIGO stage 1 AKI with a cr rise of 26umol is less likely captured than a patient requiring renal replacement therapy). This may go some way to explaining the very high mortality associated with AKI. e.g. see OAKS which was a higher risk population, with prospective data capture where AKI prevalence was 13.6% and mortality at one year was 8.6%, rather than 25% in hospital mortality reported here).

This would suggest under-recording of some of these organ injuries, meaning that many of those captured are just the most severely affected and this explains the quite high rates of in-hospital mortality associated with these organ injuries.

The exclusion/lack of information about readmissions also results in a complex selection bias; presumably those in hospital longer have greater opportunity to develop/record complications, people discharged who re-present are not captured.

I think the finding that 46% had no perioperative organ injury captured suggests a) some important injuries are missed e.g. infective, and b) those that are captured are probably an underestimate in terms of prevalence.

46% spontaneously dying without any organ injury is implausible.

2. How did the authors differentiate acute diagnoses from chronic pre-existing diseases?

For example, in code lists provided in S13; I21 is present as both AMI for the purposes of chronic 'prior MI' and for a new diagnosis of AMI.

Are there dates associated with these diagnoses or some way of differentiating a new AMI from a prior AMI? In other studies, a look back file is used to capture pre-existing diagnoses from earlier episodes of care.

This is really important because mis-classification here (primary exposure) can obviously lead to problematic interpretation.

For some conditions, surgery may be the treatment for a condition e.g. CABG for AMI, so it's important to ensure that e.g. AMI occurred after surgery to be captured as an organ injury. How was this done?

Minor comments:

- Page 8 Line 78  this is methods and should be in methods section
- Describing perioperative death as the third leading cause of death is a bit misleading because these patients all have a disease requiring surgery e.g. eg. 10% had cancer and would probably be recorded as a cancer death so there is some double counting here...
- Was death (in-hospital) time limited in any way?
- I'm not sure inclusion of Chi2 statistics in the text are helpful.
- There isn't any differentiation of emergent or elective surgery which may be helpful.

Holger K. Eltzschig, MD, PhD*Professor and Chairman**John P. and Kathrine G. McGovern Distinguished University Chair**Associate Vice President for Translational Research**Director, Center for Perioperative Medicine*Email: holger.eltzschig@uth.tmc.edu**REVIEWER COMMENTS****Reviewer #1 (Remarks to the Author):****What are the noteworthy results?**

As stated in my previous review [reviewer 1], these are certainly noteworthy given extent and size of database explored.

Thank you for your kind comments. We really hope this large population-based study on perioperative organ injury can draw the public, policymakers, clinicians, and epidemiologists' attention to this very much overlooked public health issue and help prioritize our efforts in addressing these problems.

Will the work be of significance to the field and related fields? How does it compare to the established literature? If the work is not original, please provide relevant references.

The authors have satisfactorily addressed my previous concerns- their response and discussion is measured and an exemplar of the dialogue that should happen in the review process! The issue of "under-diagnosing" is addressed, which remains a significant flaw but is now well explained to non experts and experts alike.

Thank you for your kind comments and support of this important work.

Does the work support the conclusions and claims, or is additional evidence needed?

Conclusions stack up on basis of data presented.

Thank you!

Are there any flaws in the data analysis, interpretation and conclusions? Do these prohibit publication or require revision?

All addressed from Nat Med R1.

Thank you for your kind comments.

Is the methodology sound? Does the work meet the expected standards in your field?

Yes

Is there enough detail provided in the methods for the work to be reproduced?

Code for database should be provided for other researchers to examine/explore.

Yes, codes for analysis are available to other researchers according to Nature Communications' data-sharing policies.

Reviewer #3 (Remarks to the Author):

Many thanks for the opportunity to look at this updated manuscript. The authors have addressed my concerns and performed significant additional work.

Thank you for your kind comments.

Reviewer #4 (Remarks to the Author):

Thank you for asking me to review this revised manuscript which describes an analysis of administrative care records of patients undergoing surgery in Germany between 2014-2017. The authors report the association between perioperative organ injury and in-hospital outcomes among 28M patients.

There were many excellent points raised by the first round of peer review that have been generally well addressed. I have a handful of comments based on my own experience of doing large administrative data research, the two most important are:

1. The possibility of significant information bias is not presently addressed. for example, in Table S3; 27% had Stage 1 AKI and 27% had stage 3 - this is out of keeping with existing descriptions of the distribution of AKI prevalence (e.g. OAKS study: PMID:35029656) This suggests that patients with less severe organ injury are less likely to be captured (e.g. a KDIGO stage 1 AKI with a cr rise of 26umol is less likely captured than a patient requiring renal replacement therapy). This may go some way to explaining the very high mortality associated with AKI. e.g. see OAKS which was a higher risk population, with prospective data capture where AKI prevalence was 13.6% and mortality at one year was 8.6%, rather than 25% in-hospital mortality reported here).

This would suggest under-recording of some of these organ injuries, meaning that many of those captured are just the most severely affected and this explains the quite high rates of in-hospital mortality associated with these organ injuries.

Thank you for your insightful comments and raising this important concern. We agree with the reviewer that a more severe form of organ injury might be captured more often compared with some of the mild forms in this study. Therefore, the incidence and proportion of different degrees of organ injury might be different from other studies when they have more granular data. Additionally, the impact of organ injury could be overestimated in our study due to the above information bias. As the German DRG-coding rules do not require coding AKI severity, many AKI cases have been also classified as "unspecified" in our data. We have emphasized this limitation by adding an additional paragraph in the discussion section of the manuscript. Please refer to the revised manuscript, marked version, **page 9 paragraph 3**. The reference study has also been cited in our revised manuscript.

“However, compared to some previous studies, the incidence of mild AKI might be disproportionately low to the more severe form of AKI. The overall impact of AKI on mortality might be overestimated because more severe forms of AKI are more likely captured.”

The exclusion/lack of information about readmissions also results in a complex selection bias; presumably those in hospital longer have greater opportunities to develop/record complications, people discharged who re-present are not captured.

We acknowledge that this is an intrinsic limitation of administrative data study. Our study focuses on the impact of organ injury during the specific index surgery admission. Data during re-admission will not be captured unless the patient has had another index surgery. However, upon further query of the data, only 0.01% were re-admissions from post-hospital rehabilitation facilities. Additionally, due to the differences in the healthcare system, HLOS in Germany is much longer than in the US, which allows for a more comprehensive capture of POI data. We have emphasized these selection bias in the revised manuscript, Page 12, paragraph 1.

“Similarly, selection bias, such as the impact of HLOS on the capture of perioperative organ injury, lacking of readmission data therefore to link the index surgery to perioperative organ injury that developed after hospital discharge, can not be overlooked. However, our study showed only 0.01% of the cases were re-admissions from post-hospital rehabilitation facilities. Additionally, due to the differences in the healthcare system, HLOS in Germany is much longer than in the U.S., which allows for a more comprehensive capture of POI data.”

I think the finding that 46% had no perioperative organ injury captured suggests a) some important injuries are missed e.g. infective, and b) those that are captured are probably an underestimate in terms of prevalence. 46% spontaneously dying without any organ injury is implausible.

We agree with the reviewer that the incidence of POI is likely underestimated due to the administrative nature of the data, and milder forms of organ injuries are more likely to be missed. We have added a paragraph to address the importance of other etiologies of perioperative death, such as hemorrhage and infection. Please refer to page 9, paragraph 2 of the revised manuscript.

“Our study showed about 46% of patients who died perioperatively did not experience any organ injury. The reasons could be two-fold: first, the nature of administrative data might have allowed the detection and recording of more severe forms of organ injury, and some of the milder forms of organ injury may not be recorded. Secondly, other etiologies of perioperative death, e.g., surgical complications such as acute massive hemorrhage or infection, are not captured in our study since our focus was perioperative organ injury.”

2. How did the authors differentiate acute diagnoses from chronic pre-existing diseases? For example, in code lists provided in S13; I21 is present as both AMI for the purposes of chronic 'prior MI' and for a new diagnosis of AMI. Are there dates associated with these diagnoses or some way of differentiating a new AMI from a prior AMI? In other studies, a look back file is used to capture pre-existing diagnoses from earlier episodes of care. This is really important because mis-classification here (primary exposure) can obviously lead to problematic interpretation.

For some conditions, surgery may be the treatment for a condition e.g. CABG for AMI, so it's important to ensure that e.g. AMI occurred after surgery to be captured as an organ injury. How was this done?

Thank you for this insightful comment. We have performed extensive revisions of the statistical analysis to address your concern. Specifically, we have double-checked POI and comorbidities for overlapping codes and

corrected overlapping diagnoses: any ICD code, including subcodes, is now mutually exclusive in either a POI or a comorbidity category. In detail:

- Specified K7040 and K7042 as POI acute liver injury (before K704 in POI was overlapping with K704 in comorbidity moderate/severe liver disease) and excluded these from comorbidity moderate/severe liver disease
- Specified K712 as POI acute liver injury because K71 was too broad and partially overlapping with comorbidity mild liver disease
- Deleted F051 from comorbidity dementia because this was overlapping with POI delirium
- Deleted I63 and I64 from comorbidity cerebrovascular disease because this was overlapping with POI stroke
- Deleted I21 and I22 from comorbidity Myocardial infarction because this was overlapping with POI acute myocardial infarction

After making the abovementioned changes, we re-analyzed the entire study and updated all tables and figures throughout the manuscript, including supplementary documents. Please refer to the revised manuscript, **Table 1,2,3, and Figure 1; then Table S 1-11, and 13**. We are delighted that the results have not largely changed (only some minor points difference), except that now ALI is associated with the highest incidence of in-hospital death. To highlight its impact, we have now also included a paragraph in the manuscript discussion section; please refer to the revised manuscript, marked version, **page 10, and paragraph 2**.

“Our study showed perioperative ALI is rare (0.1%) but associated with the highest mortality (mortality: 68.7%). Compared to AKI, Perioperative ALI is a much less investigated perioperative organ injury. Mild liver dysfunction in patients without liver disease is common following major surgery. However, acute “ischemic hepatitis,” a diffuse injury that arises secondary to hypoperfusion (hemodynamic instability) and hypoxia, and is exacerbated by systemic inflammation, could be associated with significant perioperative mortality and morbidity, especially in patients with preexisting liver disease and cardiac dysfunction. Preventive and therapeutic measures for perioperative ALI are urgently needed.”

Additionally, to address the limitation of lack of timeline granularity, we have added the following in the discussion of the revised manuscript (**Page 12, paragraph 1**).

“This lack of granularity may also affect our ability to determine the timeline of organ injury in contrast to the index surgery and differentiate pre-existing comorbidities from organ injury after surgery. However, our analysis was performed diligently to ensure that all ICD-codes, including subcodes, are mutually exclusive in either a perioperative organ injury or comorbidity category.”

Minor comments:

- Page 8 Line 78  this is methods and should be in methods section

This has been revised accordingly. Please refer to **Page 25, paragraph 2** of the revised manuscript, marked version.

- Describing perioperative death as the third leading cause of death is a bit misleading because these patients all have a disease requiring surgery e.g. eg. 10% had cancer and would probably be recorded as a cancer death so there is some double counting here...

Thank you for this insightful comment. We agree that the etiology of death could overlap and double counting exists. For example, if a patient went to surgery for cancer and had an acute myocardial infarction after surgery and died, very likely, this patient will be counted as a case that died from cancer, cardiovascular disease, and perioperative death. To highlight this, we have added an explanation sentence in Page 8, paragraph 2.

“If perioperative death could be considered a single separate disease entity regardless of the causes for surgery (such as cancer and acute myocardial infarction that could have led to surgery and the patient could die perioperatively from cancer or acute myocardial infarction), this would account for the third leading cause of death in Germany, following circulatory system diseases and neoplasms.”

- Was death (in-hospital) time limited in any way?

The data presented in our manuscript is in-hospital death; no specific timeline is selected. As a significant amount of deaths occurred after hospital discharge, one-year mortality would be a more comprehensive assessment of the impact of perioperative organ injury. This limitation was highlighted in the limitation of the study, discussion section. Please refer to Page 12, paragraph 1.

“Lastly, multiple studies have shown that in-hospital mortality is an incomplete measure of surgical mortality, since a significant percentage of procedural deaths occur after transfer or discharge from the index hospital. Since all the patient data are de-identified before being sent to the federal statistics office, we cannot link these data/patients with a death record. Therefore, we cannot provide one-year mortality data. Examining the one-year mortality would be a more comprehensive assessment of the impact of perioperative organ injury, as has been used in other studies.”

- I'm not sure inclusion of Chi2 statistics in the text are helpful.

Thank you! We have removed it accordingly.

- There isn't any differentiation of emergent or elective surgery which may be helpful.

Thank you for this important comment. Unfortunately, our data does not include elective or emergency surgery. Our data only have the data on emergency admission (more patients in the POI group had emergent admission compared with non-POI group patients, 54.7% vs. 28.1%). Investigating the role of emergency surgery on POI and surgical outcomes in this large database would be very insightful as some literature shows that emergent surgery is a significant risk factor for poorer patient outcomes (PMID: 37133876, 37714262). We emphasized this limitation in our discussion section, please refer to page 12, paragraph 1.

“Similarly, the impact of emergency surgery vs. elective surgery could not be determined in our study due to lack of such data, but investigating the role of emergency surgery on POI and surgical outcomes in this large database would be very insightful as some literature shows that emergent surgery is a significant risk factor for poorer patient outcomes.”